# Influence of Lemongrass Essential Oil (*Cymbopogon flexuosus*) Supplementation on Diabetes in Rat Model

**DOI:** 10.3390/life14030336

**Published:** 2024-03-04

**Authors:** Ailton S. S. Júnior, Felipe J. Aidar, Luiz A. S. Silva, Thais de B. Silva, Sara F. M. de Almeida, Daiane C. S. Teles, Waldecy de L. Junior, Dulce M. Schimieguel, Daniel A. de Souza, Ana C. S. Nascimento, Enilton A. Camargo, Jymmys L. dos Santos, Ana M. de O. e Silva, Rogéria de S. Nunes, Lysandro P. Borges, Ana A. M. Lira

**Affiliations:** 1Department of Pharmaceutical Sciences, Federal University of Sergipe, São Cristóvão 49100-000, Sergipe, Brazilluizandradessilva@yahoo.com.br (L.A.S.S.); thay_britto@hotmail.com (T.d.B.S.); sarinha_sfma@hotmail.com (S.F.M.d.A.); daianecarolinest@gmail.com (D.C.S.T.); dulcemarta@hotmail.com (D.M.S.); rogeriafarm@academico.ufs.br (R.d.S.N.); ana_lira2@hotmail.com (A.A.M.L.); 2Department of Physical Education, Federal University of Sergipe, São Cristóvão 49100-000, Sergipe, Brazil; fjaidar@gmail.com (F.J.A.); jymmyslopes@yahoo.com.br (J.L.d.S.); 3Department of Morphology, Federal University of Sergipe, São Cristóvão 49100-000, Sergipe, Brazil; wluccajr@academico.ufs.br; 4Department of Physiology, Federal University of Sergipe, São Cristóvão 49100-000, Sergipe, Brazil; dani.nutriufs@gmail.com (D.A.d.S.); nutrianacarlanascimento@gmail.com (A.C.S.N.);; 5Department of Nutrition, Federal University of Sergipe, São Cristóvão 49100-000, Sergipe, Brazil; anamaraufs@gmail.com; 6Department of Immunology, Institute of Biomedical Sciences, University of São Paulo, São Paulo 05508-220, São Paulo, Brazil

**Keywords:** *Cymbopogon*, citral, oxidative stress, hypoglycemic agents, diabetes mellitus

## Abstract

(1) Background: Species of the genus *Cymbopogon* and its essential oil are known for their antioxidant and hypoglycemic effects. This study aimed to investigate the impact of the essential oil of *Cymbopogon flexuosus* (EOCF), and its major component, citral, on glycemic, lipid, antioxidant parameters, and oxidative stress in a type 1 diabetes (DM1) rat model. (2) Methods: Initially, EOCF was analyzed by Gas chromatography-mass spectrometry (GC-MS) and the antioxidant activity of EOCF and citral was evaluated. Next, male Wistar rats (3 months old, 200–250 g) induced with DM1 using Streptozotocin (STZ) were divided into four groups: negative control supplemented with an 80% Tween solution, two groups of animals supplemented with EOCF (32 mg/kg and 64 mg/kg) and with citral (32 mg/kg), and treated for 14 days. Measurements of blood glucose levels and body weight were taken; after euthanasia, biochemical markers, including lipid profile, uric acid, alanine aminotransferase (ALT), and aspartate aminotransferase (AST), were evaluated. (3) Results: The predominant compounds in EOCF were α-citral (53.21%) and neral (19.42%), constituting 72.63% citral. EOCF showed good antioxidant activity, significantly greater than citral. EOCF supplementation demonstrated a mitigating effect on glycemic, lipid, and hepatic abnormalities induced by DM1. (4) Conclusions: EOCF emerges as a promising therapeutic option for the management of DM1.

## 1. Introduction

Diabetes mellitus (DM) is a metabolic disorder that has severe global health impacts [1]. Currently, approximately 537 million adults (20–79 years old) worldwide have diabetes, and by 2045, this number is estimated to increase to around 783 million, confirming diabetes as a significant global health challenge [2].

International organizations, such as the American Diabetes Association, classify diabetes based on its etiopathogenesis into four categories, including type 1 diabetes mellitus (DM1), type 2 diabetes mellitus (DM2), gestational diabetes mellitus (GDM) and other types [3]. Among these types of diabetes, DM1 has been extensively studied. In 2022, data show that out of 8.75 million people with diabetes, only 1.52 million were under 20 years old, and 62% of new DM1 cases occurred in individuals over 20 years old [2].

DM1 is characterized by the residual or absent production of insulin, resulting from the destruction of beta cells, related to autoimmunity, with the production of reactive oxygen species (ROS) at high concentrations [3,4,5,6]. Therefore, if not diagnosed early and if there is no proper treatment, DM1 can lead to rapid mortality. Raising awareness of the condition among the population is crucial [2].

At present, managing patients with this condition encompasses diverse therapeutic strategies, with a predominant focus on pharmacological intervention. This approach involves the administration of hypoglycemic agents, either as standalone treatments or in conjunction with exogenous insulin when the optimal control of glucose and glycated hemoglobin (HbA1c) levels cannot be achieved [7,8]. Emphasizing the significance of time in range, a key aspect of diabetes care, becomes integral to assessing and refining the effectiveness of these interventions.

However, despite the therapeutic benefits of the mentioned methods, limitations are evident in the literature, prompting the search for new therapeutic approaches. In this regard, alternative treatments, especially herbal medicines, have gained significant relevance in current discussions [9,10,11].

From this perspective, research has identified some promising plants for glycemic control in diabetic patients [12,13,14]. Among these, the essential oils of the *Cymbopogon* genus, rich in citral, stand out for their significant antidiabetic and antioxidant properties in previous studies [15,16]. However, studies using the essential oil of *Cymbopogon flexuosus* (EOCF) in this area are scarce in the literature [17,18,19,20,21,22,23,24,25].

Therefore, with an innovative approach, the aim of this study is to investigate the antioxidant, hypoglycemic, anti-dyslipidemic and oxidative stress effects of EOCF in Wistar rats induced with streptozotocin (STZ) to develop DM1.

## 2. Materials and Methods

### 2.1. Essential Oil

The EOCF was commercially obtained from local suppliers (Engenharia das Essências, lot: 451 A225841), produced by Yanih Cosmetics—Brazilian Industry, ANVISA Notification 25351.25600/2017-36, with 100% purity, as outlined in Annex 1. The isolated compound citral was acquired from Sigma Aldrich (St. Louis, MO, USA) with a purity content of 95%.

### 2.2. Gas Chromatography-Mass Spectrometry (GC-MS)

The EOCF underwent GC-MS analysis using a Thermo Scientific gas chromatograph, Bremen, Germany, model TRACE 1310 and a mass spectrometer model TSQ-9000, with the TriPlus RSH autosampler. An NA-5MS column (60 m × 0.25 mm ID, 0.25-μm film thickness) was employed for compound separation, with helium as the carrier gas at 99.999% purity (White Martins S.A) at a flow rate of 1 mL/min and a split/splitless autoinjector.

For the analysis, an essential oil solution was prepared at a concentration of approximately 10 mg/mL using hexane as the solvent, measured in a 1.5 mL glass vial. Subsequently, the sample was submitted for chromatographic analysis. The ramp programming was as follows: 60 °C–3 °C/min–240 °C (15 min). The injection mode was split (1:30), in SCAN mode, with a total analysis time of 75 min. For MS, the conditions were as follows: injector temperature 220 °C, detector temperature 240 °C, solvent cutoff time 2 min, electron ionization (EI) mode at 70 eV with a mass-to-charge ratio (*m*/*z*) range of 40 to 350 Daltons. The identification of essential oil components was carried out through their retention indices (RI), calculated for each constituent by injecting a series of linear hydrocarbon standards (C8–C20) under the same sample conditions and compared with tabulated values, confirming the identification by comparing the compound spectra with the reference, presented by the Nist 107, 21 and Wiley 8 libraries.

### 2.3. Determination of Antioxidant Activity

The antioxidant activity of EOCF and citral was assessed using the compost 1,1-difenil-2-picril-hidrazila (DPPH) radical assay by the methodology adapted from Brand-Williams et al. [17] with some modifications; 50 μL of EOCF, citral (solubilized in ethanol) and trolox (positive control—concentration of 50 μg/mL) were transferred. Subsequently, 150 μL of methanolic DPPH solution (0.6 mM/L) was added. After 30 min, the absorbance was read at 515 nm. The DPPH radical scavenging activity was expressed as a percentage calculated from the absorbance values of the control and the sample.

The antioxidant activity of EOCF and citral was also assessed using the radical monocation of 2,2′-azinobis-(3-ethylbenzothiazoline-6-sulfonic acid) (ABTS). EOCF and citral were dissolved in ethanol and the samples were analyzed by the methodology adapted from [17]. ABTS was activated 16 h before the experiment by mixing 1.25 mL of ABTS stock solution and 22 μL of potassium persulfate solution; 30 μL of EOCF, citral (solubilized in ethanol) and trolox (positive control—concentration 50 μg/mL) were pipetted in. Subsequently, 300 μL of ABTS stock solution was added. After 6 min, it was read in a spectrophotometer at a wavelength of 734 nm. The ABTS radical scavenging activity was expressed as a percentage calculated from the absorbance of the control and the sample. Antioxidant activity was also determined using the Ferric Reducing Antioxidant Power (FRAP) method, following Oyaizu’s method [18] with some modifications. In short, 9 μL of samples, 27 μL of water and 270 μL of FRAP reagent (a mixture of ferric chloride, TPTZ and acetate buffer (0.3 M, pH 3.6). The plate was incubated at 37 °C for 30 min, and absorbance at 595 nm was recorded using a plate reader, with the results expressed in absorbance.

### 2.4. Animals

Male Wistar rats, aged 3 months and weighing approximately 250–300 g, were used in the study. These rats were obtained from the Physiology Sector Animal Facility at the Federal University of Sergipe (UFS), São Cristóvão, Sergipe, Brazil. Twenty animals were randomly housed in appropriate cages under controlled temperature conditions (22 ± 3 °C) with a 12-h light/dark cycle (lights on from 06:00 to 18:00), providing 300 lux of light. They had free access to specific rodent feed (Labina^®^) and water ad libitum. The procedures carried out in this study were approved by the Ethics Committee on Animal Use at UFS, under CEUA No. 8502100821, on 24 November 2021.

### 2.5. Induction of Diabetes Mellitus

Experimental DM1 was induced as described by Wang et al. [26] and Barman et al. [27], using the drug STZ at a dose of 40 mg/kg of body weight, dissolved in 0.01 M citrate buffer, pH 4.5 and administered intraperitoneally. The STZ administration was carried out, and 30 min later, all groups were provided with food to prevent hypoglycemia. Blood was collected through tail puncture for blood glucose measurement using an Accu-Chek Go glucometer (Roche Diagnostics GmbH, D-68298, Mannheim, Germany) 72 h and 96 h after induction. Only animals with fasting blood glucose equal to or greater than 150 mg/dL were included in the study [28].

After confirming the induction of diabetes, the animals were randomly assigned and treatment was started with a total of 20 animals.

During the experimental period, the animals were observed daily in cages for mortality or signs of toxic effects. At the end of the experiment, there was no mortality in any of the groups.

### 2.6. Experimental Groups

Body weight and blood glucose measurements (tail puncture) were properly assessed on days 0, 4, 8 and 14. The treatment occurred daily for 14 days. The animals received treatment via intragastric gavage and were divided into four groups (five animals/group), namely: (1) Control (Tween 80—TW 80): diabetic animals treated with the vehicle (0.2% saline solution + Tween 80); (2) EOCF 32 (32): diabetic animals treated with the EOCF supplementation solution at a dose of 32 mg/kg of body weight; (3) EOCF 64 (64): diabetic animals treated with the EOCF supplementation solution at a dose of 64 mg/kg of body weight; (4) Citral (CT): diabetic animals treated with citral supplementation at a dose of 32 mg/kg of body weight, as shown in Table 1.

To disperse the essential oil at doses of 32 and 64 mg/kg, 0.2% of the surfactant Polysorbate (Tween 80) was used along with the saline solution and agitated in a vortex immediately before administration. Thus, for the control group, Tween 80 (0.2%) and saline solution were also used to ensure standardization and minimize the risk of study bias.

The assay with experimental animals followed the standards established by the National Council for the Control of Animal Experimentation (CONCEA) and the Ethics Committee on Animal Use (CEUA) of the Federal University of Sergipe (UFS), complying with the minimum number of animals required for use in pre-clinical assays [29,30].

### 2.7. Supplementation

The supplementation involved the administration of EOCF and the isolated compound citral dispersed in TW 80. EOCF was administered at doses of 32 mg/kg and 64 mg/kg of body weight, while the recommended dose for citral in the literature, 32 mg/kg [31], was used. The administration was carried out daily, at the same time each day, via intragastric gavage, using a specific stainless steel cannula for rodents with a rounded tip to prevent injury.

### 2.8. Sample Collection

After 14 days, the animals were euthanized following the administration of a combination of ketamine (100 mg/kg) and xylazine (10 mg/kg) via intraperitoneal injection. Blood and tissues (pancreas, spleen, liver and kidneys) were collected, weighed and stored for further analysis.

### 2.9. Determination of Serum Biochemical Markers

Blood was centrifuged at 800× *g* for 15 min at 4 °C, and serum was stored at −80 °C. Serum concentrations of triglycerides (TG), total cholesterol (TC), HDL-Cholesterol (HDL-C), alanine aminotransferase (ALT), aspartate aminotransferase (AST), creatine kinase (CK), lactate dehydrogenase (LDH) and uric acid were determined according to the manufacturer’s procedures (Labtest^®^, Lagoa Santa, Minas Gerais, Brazil).

### 2.10. Determination of Oxidative Stress Markers in Tissues

The organs were removed and washed three times in a potassium chloride solution (KCl 1.15%) during homogenate preparation. Subsequently, they were homogenized (1:5 *w*/*v*) with a solution of KCl, phenylmethylsulfonyl fluoride (PMSF 100 m.mol^−1^) and Triton solution (10%). The homogenates were then centrifuged at 3000× *g* for 10 min at 4 °C, and the supernatant was stored at −80 °C for the analysis of oxidative stress markers (TBARS) following the methodology described in the previous section. The results were expressed per gram of tissue.

### 2.11. Statistical Analysis

All statistical analyses were conducted using Graph Pad Prism version 5.0 and presented as means ± standard deviation (X ± SD) from quintuplicate (for antioxidant samples) and quintuplicate (for biochemical samples) analyses. First, data were assessed for normality using the Shapiro–Wilk test and then statistically analyzed between groups using one-way analysis of variance (ANOVA) and Tukey post-hoc tests. Statistically significant differences between the adopted samples were considered when *p* < 0.05.

## 3. Results

### 3.1. Chemical Composition of EOCF

The results obtained from the GC-MS for the analysis of EOCF, are presented in Table 2. Thirteen compounds were identified, constituting a total sample value of 94.87%. The major components detected were α-citral (C_10_H_16)_, representing 53.21% of the area, followed by Neral (C_10_H_16_O) at 19.42% and 12.58% Geraniol (C_10_H_18_). EOCF also contained smaller percentages of constituents such astricyclo, α-pinene, caryophyllene, nonanone, linalool, isogeraniol, isoneral, isocitral, caryophyllene, and geranyl acetate, present in lower proportions ranging from 0.07 to 4.90%.

### 3.2. Antioxidant Activity of EOCF

The antioxidant action can be assessed through various methods utilizing different mechanisms, such as metal chelation and electron transfer. Classic examples of these techniques include the measurement of DPPH and ABTS radical scavenging and the FRAP assay. The antioxidant action of EOCF and citral based on these methods can be observed in Figure 1.

#### 3.2.1. DPPH Radical Scavenging

In Figure 1A, the findings from the radical scavenging tests are presented. The EOCF sample at a concentration of 100 mg/mL resulted in superior antioxidant activities compared to the control system (DPPH without the antioxidant) and the citral group (*p* < 0.0001).

It is observed that EOCF exhibited higher DPPH radical scavenging activity (86.45%), while citral inhibited the activity by 50.72%, and Trolox by 43.65%. Additionally, EOCF and citral showed statistical differences when compared to the control system.

#### 3.2.2. ABTS Radical Scavenging

As with the DPPH reduction method, the ABTS capture method uses a synthetic radical. The ABTS radical capture method is similar to the DPPH method. The results of the research for the EOCF and citral samples are shown in Figure 1B. Although EOCF and citral showed significant differences when compared to the control system (*p* < 0.0001), the inhibition of ABTS activity by EOCF proved to be more efficient, with a statistically significant difference observed between EOCF and citral (*p* < 0.0001). Therefore, it can be said that the antioxidant potential of EOCF was greater (75.73%) than that of Trolox (64.83%) and citral (29.55%).

#### 3.2.3. FRAP (Ferric Reducing Antioxidant Power)

According to Figure 1C, significant differences (*p* < 0.01) were found between the FRAP activities of EOCF, citral and the system (FRAP reagent without a sample). In contrast, the FRAP activities for EOCF and citral exhibited significant differences (*p* < 0.001). EOCF resulted in a higher Fe^3+^ reducing potential than isolated citral.

### 3.3. Anti Hyperglycemic Activity

A significant increase in blood glucose levels was observed in control diabetic rats induced by STZ compared to supplemented animals (Figure 2A). Treatment with citral and EOCF significantly corrected this increase when compared to the control (TW 80); however, the higher dose (64 mg/kg) did not have a significant effect compared to the lower dose (32 mg/kg), and there were no significant differences between EOCF and citral. However, weight gain in STZ-induced diabetic rats treated with EOCF and citral supplementation compared to control diabetic animals showed no significant differences (Figure 2B).

### 3.4. Effect of EOCF on Hepatic and Renal Functions

The exposure of rats to STZ resulted in hepatic dysfunctions, as indicated by ALT and AST activities (Figure 3A,B). Regarding renal function, we demonstrated that exposure to STZ induced an increase in uric acid levels (Figure 3C). Treatment with EOCF (32 and 64 mg/kg, gavage) and citral significantly protected against hepatic ALT dysfunctions, as shown in Figure 3A. However, AST (Figure 3B) exhibited significant differences only with the higher dose and citral when compared to the control (TW 80). Regarding the renal marker, we did not observe significant differences, as shown in Figure 3C.

### 3.5. Effect of EOCF on Metabolic Lipid Parameters

Figure 4A–D demonstrated that the exposure of rats to STZ induced significant metabolic lipid disturbances. A significant increase in plasma concentrations of Triglyceride (TG), Total cholesterol (TC) and LDL-Cholesterol (LDL), along with a decrease in HDL-Cholesterol (HDL-C) levels, was observed in the control group of diabetic rats. Citral (alone) only showed a statistical difference in HDL-C and did not significantly alter the other metabolic parameters.

## 4. Discussion

In this study, the antioxidant activity and the effects of the oral supplementation with EOCF in diabetic rats were evaluated.

The predominant chemical constituents of EOCF are the aldehydes neral and geranial, which when combined, form citral. However, their composition can vary due to various factors. In our study, the citral values (72.63 α-citral + neral) were similar to other studies in the literature. On the other hand, the citral value in our study was higher than in the studies of Anaruma et al. [32] and Vera et al. [33]. The primary constituents in our findings were α-citral, followed by neral and geraniol. Other studies also highlight citral as the primary component of *Cymbopogon flexuosus* and its essential oil. The third major constituent in our findings was geraniol; a study by Mata et al. [34] described geraniol as a potent antioxidant. Ganjewala et al. [35] also reported that many of the properties of lemongrass oil are associated with compounds like citral and geraniol.

The differences in the chemical composition of EOCF in this research can be justified by geographical origin, plant development, harvest season and climate, as well as the method used for extraction [36].

The antioxidant activity of EOCF in this research was also evident. Antioxidant constituents can be found in various plants and have a mitigating action on oxidation by one or more mechanisms, which can prevent oxidative stress. These constituents include free radical scavengers, reducing agents, and chelating agents, among others [37,38]. Various essential oils have been extensively researched for their antioxidant activity [39].

Regarding the DPPH antioxidant activity, EOCF demonstrated a high antioxidant capacity when compared to citral and the control system. Our results show that EOCF effectively eliminates free radicals, making it a potent antioxidant [10].

This study aimed to assess the antidiabetic activity of EOCF and its effects on glycemic control, lipid and hepatic parameters in STZ-induced type 1 diabetic rats. STZ is widely used to induce experimental diabetes in rats, leading to conditions similar to those in human diabetic patients. Symptoms include body weight reduction, polyuria, polydipsia and polyphagia, which persisted throughout the experiment.

All the groups of animals experienced body weight loss over the course of the experiment. This is associated with the chronic implications of type 1 diabetes, indicating lipid and protein catabolism, accompanied by polyuria and glucosuria [40,41]. Rats supplemented with EOCF at doses of 32 and 64 mg/kg and citral did not attenuate weight loss when compared to the control group (TW 80). This might be due to the relatively short intervention period of 14 days. Further studies should monitor body weight over an extended period and analyze body fat to understand whether the reduction in body weight is due to STZ induction or the action of STZ and whether *C. flexuosus* supplementation can attenuate or accelerate body weight loss.

Glycemic disturbances in the long term lead to various metabolic changes, including non-ketotic hyperosmolar syndrome, peripheral autonomic neuropathy, retinopathy, and ketoacidosis [42,43,44].

In diseases like diabetes, glucose homeostasis imbalance is a significant factor in disease progression. In diabetes, insulin deficiency reduces the activities of crucial enzymes in glycolysis, such as glucokinase, pyruvate kinase, and hexokinase, while elevating glucose-6-phosphatase, crucial for gluconeogenesis [45]. This results in hyperglycemia and contributes to a reduction in tissue and cellular energy substrates, leading to an increased formation of reactive oxygen species (ROS) and ATP synthesis [46,47,48,49].

In this context, it is essential for type 1 diabetic patients to control plasma glucose levels close to ideal values, as maintaining blood glucose levels within the ideal range is critical to reducing the metabolic complications associated with diabetes [50,51].

Citral, a primary aliphatic monoterpene aldehyde in various *Cymbopogon* species, is composed of the enantiomers geranial (trans) and neral (cis) [52]. In this study, oral supplementation with EOCF at both doses and with citral in type 1 diabetic rats for 14 days significantly reduced blood glucose levels. This effect aligns with other studies that used citral alone or aqueous extracts of *Cymbopogon citratus* in different doses [31,53,54,55]. Supplementation with EOCF also reduced hyperlipidemia in diabetic rats, showing reductions in TC and triglycerides, as well as an increase in HDL-C levels. In our study, EOCF supplementation demonstrated a higher lipid profile improvement in diabetic animals, with reduced TC, triglycerides, and increased HDL-C.

Chronic metabolic changes with fluctuations in the elevation of reactive species and/or antioxidant activities are associated with oxidative stress, which is implicit in various pathologies, including diabetes [56].

In DM, the homeostatic imbalance of glucose is a determining factor in the progression of the disease. This deregulation causes the activation of reactive oxygen species (ROS) forming pathways and suppression of ATP genesis [47,48,49]. Thus, the use of antioxidant compounds has positive effects on the redox effect, reducing damage to biochemical parameters and some tissues [57,58]. Therefore, in our study, the supplementation of EOCF and citral showed protective effects against the damage caused by reactive oxygen species produced by DM.

This study demonstrated that experimental type 1 diabetes also induced tissue damage, as evidenced by a significant increase in ALT, AST and uric acid levels in diabetic animals. Supplementation with EOCF at doses of 32 mg/kg significantly reduced ALT and AST markers compared to the control group. However, there was no significant difference when comparing with the group that received isolated citral. The group that received 64 mg/kg EOCF significantly reduced the AST marker but showed no significant difference compared to citral. There were no differences in uric acid concentrations between the animals supplemented with EOCF or citral compared to the control group.

Overall, the study suggests that EOCF and citral supplementation can be effective in reducing oxidative stress and improving glycemic, lipid and hepatic parameters. Considering the scarcity of studies available using *Cymbopogon* species, the use of *C. flexuosus* in this research is pioneering, and our results indicate a higher antioxidant activity compared to citral alone. This essential oil also showed promise in improving the glycemic, lipid, and hepatic profiles in diabetic Wistar rats.

## 5. Conclusions

The use of EOCF demonstrated efficacy in ameliorating the chronic metabolic alterations associated with type 1 diabetes (DM1), specifically influencing glycemic, lipid, and hepatic parameters. The observed effects suggest a potential therapeutic avenue, with mechanisms of action linked to a reduction in reactive species markers and notable antioxidant activity, attributable to the synergistic interaction among major and minor compounds. This highlights the prospect of oral lemongrass essential oil supplementation as a viable non-pharmacological or pharmacological option in DM1 treatment.

In light of the obtained results, our hypothesis posits that daily oral supplementation with EOCF, even in the absence of insulin administration for DM1, may instigate metabolic regulations leading to a moderation of blood glucose levels, the control of systemic inflammatory activity, and regulation of immune cell recruitment, all of which are consequences of uncontrolled diabetes. The intriguing concept of insulin-independent glucose regulation suggests a novel avenue for exploration, potentially mediated through reactive oxygen species-dependent mechanisms [57,58]. However, it is imperative to acknowledge that these findings, along with subsequent investigations, necessitate comprehensive exploration to ascertain the optimal dosage and safety profile of antioxidant supplementation derived from the isolated essential oil, either alone or when incorporated into new formulations, and to comprehensively understand its systemic effects. It is also necessary to conduct experimental studies using standard drug therapies (antidiabetics) in a positive control group in order to highlight the clinical significance of OECF and its identified constituents in terms of relevance.

## Figures and Tables

**Figure 1 life-14-00336-f001:**
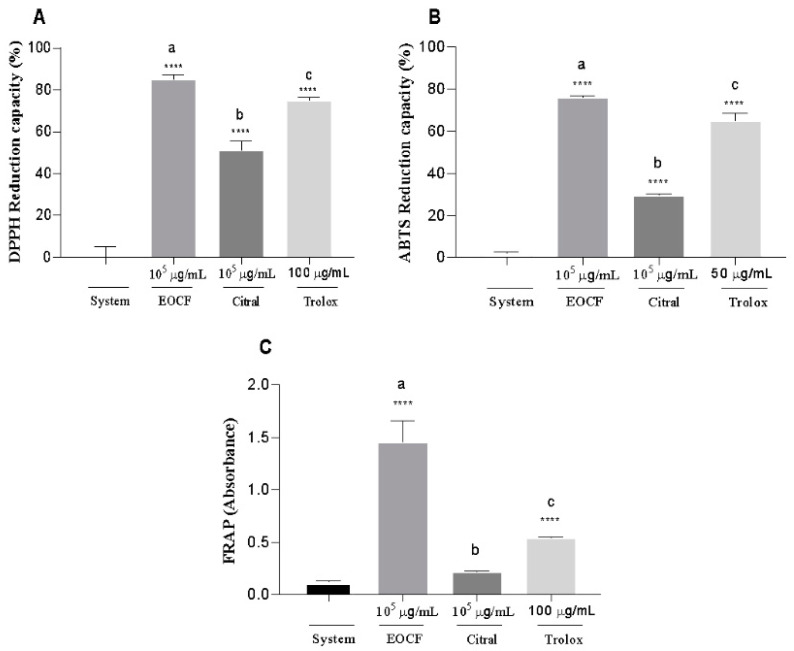
Antioxidant activity of *Cymbopogon flexuosus* essential oil (EOCF) and citral. The analysis was carried out using the (**A**) DPPH, (**B**) ABTS and (**C**) FRAP methods. Trolox was used as the standard antioxidant and the system was used as a control. The test was carried out in quintuplicate and the results represent the mean ± standard deviation (SD) of the values; different letters indicate a statistical difference (*p* < 0.05) between the samples tested; **** indicates a statistical difference between the samples and the system (without antioxidant) *p* < 0.05—(ANOVA followed by Tukey’s post-test).

**Figure 2 life-14-00336-f002:**
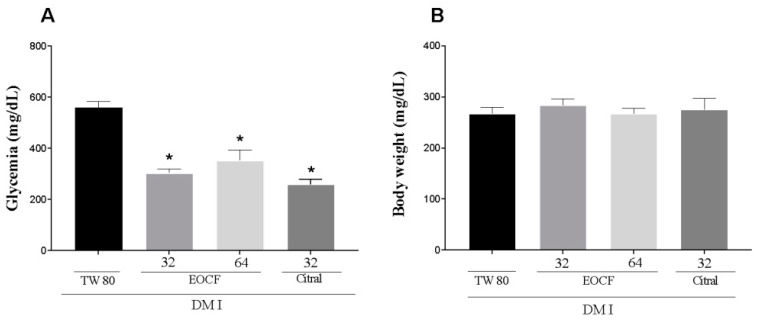
Glycemia (**A**) and body weight (**B**) of diabetic rats after 14 days of supplementation with EOCF (32 and 64 mg/kg) and citral (32 mg/kg). Tween 80 (TW80) was used as a control. The assay was performed in quintuplicate and the results represent the mean ± standard deviation (SD) of the values; * *p* < 0.05 versus system—(ANOVA followed by Tukey post-test).

**Figure 3 life-14-00336-f003:**
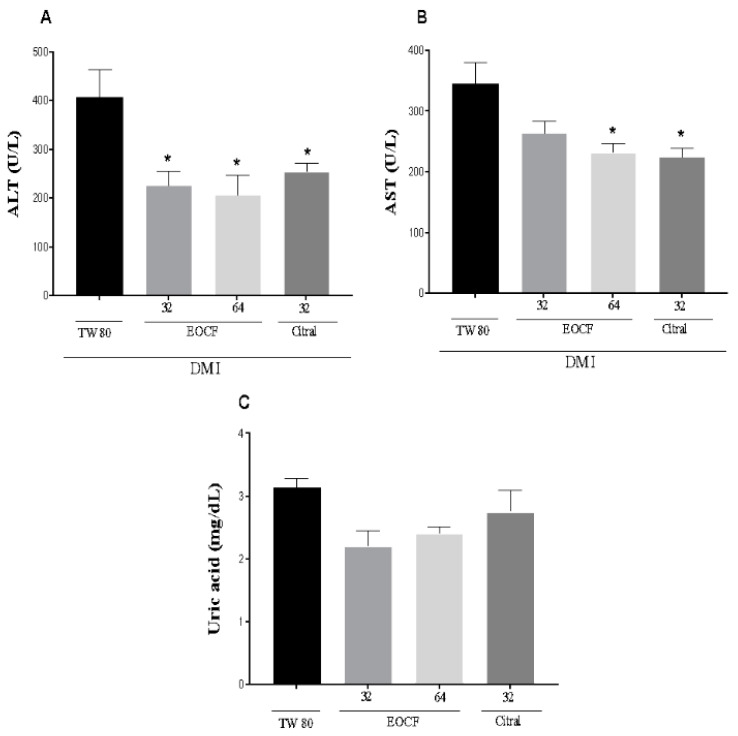
Liver ALT (**A**), AST (**B**) and Uric Acid Levels (**C**) of diabetic rats after 14 days of supplementation with EOCF (32 and 64 mg/kg) and citral (32 mg/kg). Tween 80 (TW80) was used as a control. The assay was performed in quintuplicate and the results represent the mean ± standard deviation (SD) of the values; * *p* < 0.05 versus system—(ANOVA followed by Tukey post-test).

**Figure 4 life-14-00336-f004:**
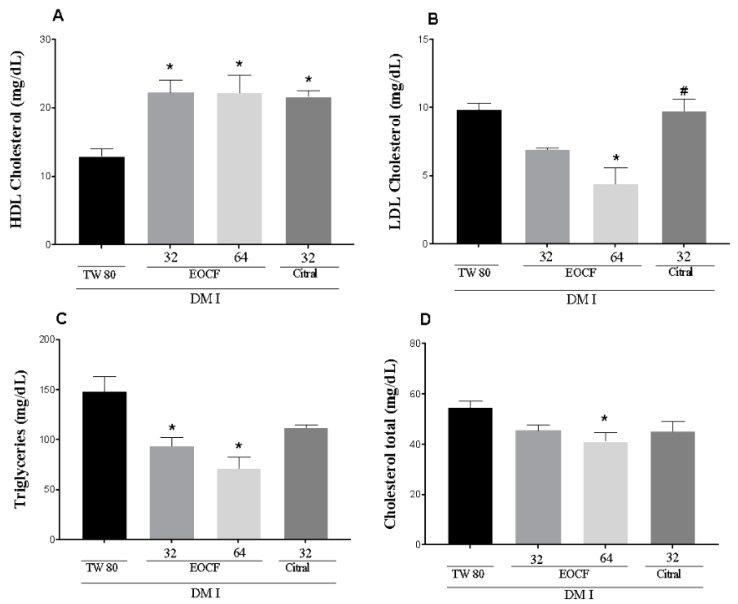
Lipid levels of HDL-Cholesterol (**A**), LDL-Cholesterol (**B**), Triglyceride (**C**) and Total cholesterol (**D**) of diabetic rats after 14 days of supplementation with EOCF (32 and 64 mg/kg) and citral (32 mg/kg). Tween 80 (TW80) was used as a control. The assay was performed in quintuplicate and the results represent the mean ± standard deviation (SD) of the values; * *p* < 0.05 versus system, # *p* < 0.05 versus EOCF 64 mg/kg and citral—(ANOVA followed by Tukey post-test).

**Table 1 life-14-00336-t001:** Experimental groups of the in vivo study.

Experimental Group	Species	Number of Groups	Initial Number of Animals/Final Numbers of Animals	Total Number
TW 80	Rats *Wistar*	1	5/5	5
EOCF 32	Rats *Wistar*	1	5/5	5
EOCF 64	Rats *Wistar*	1	5/5	5
CT	Rats *Wistar*	1	5/5	5
**Total**:				20

**Table 2 life-14-00336-t002:** Chemical constituents of lemongrass essential oil. IR lit.: literary retention index, IR exp.: experimental retention index.

Peak	Time of Retention (min)	Compound	Experimental IR	Literary IR	Relative Area (%)
1	10.61	Tricyclene	928	926	0.07
2	10.98	Pinene <α->	937	939	0.11
3	11.69	Camphene	956	954	0.64
4	16.67	Nonanone	1074	1090	0.52
5	17.96	Linalool	1103	1096	0.65
6	20.50	Isogeraniol	1158	1229	0.30
7	20.88	Isoneral	1166	1164	0.38
8	21.76	Isocitral	1185	1180	0.74
9	24.66	Neral	1248	1238	19.42
10	24.99	Geraniol	1256	1252	12.58
11	26.02	α-citral (3,7-dimethyl-2,6-octadienal)	1278	1318	53.21
12	30.63	Geranyl acetate	1381	1381	4.90
13	32.82	Caryophyllene	1432	1419	1.35

## Data Availability

The datasets used and/or analyzed during the current study are available from the corresponding author upon reasonable request.

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
