# Peer review of "Influence of Lemongrass Essential Oil (Cymbopogon flexuosus) Supplementation on Diabetes in Rat Model"

_life, 2024, doi:10.3390/life14030336_

Round 1

Reviewer 1 Report

Comments and Suggestions for Authors

REVIEWER’S COMMENTS

for:

Manuscript entitledInfluence of Lemongrass Essential Oil (Cymbopogon flexuosus) supplementation on Diabetes in Rat Models”

Authors: Ailton S. S. Júnior et al.

Journal: Life; Manuscript ID:

Summary:

In this article Ailton S. S. Júnior, and colleagues describe the clinical significance of the essential oil content of lemongrass in relation to type 1 diabetes mellitus and its potential effects on lipid homeostasis. The overall topic is intriguing, and this manuscript aims to expand opportunities in the treatment of diabetes mellitus, a global health challenge.

General comments

Overall, the reviewer is satisfied with the composition of the manuscript, as it describes relevant clinical evidence with potential benefits for both physicians and patients. The scientific English used in the manuscript is acceptable, though proofreading by a native English speaker would enhance its quality. A revision of the manuscript is necessary. Please read my major and minor comments below/above.

Major comments

1.     In the title of the article the “Rat Models” are in Plural Form, which indicates that there will be more than one model mentioned in the article. Since the authors describe only the streptozotocin-induced model of diabetes mellitus, please correct this grammatical mistake.

2.     There are numerous mistakes in the Abstract section which indicates it to be rewritten. Based upon this reviewer’s understanding (according to the GC-MS examination’s results), the main active component of the EOCF is citral, which is responsible for the positive effects demonstrated in the Results part of the manuscript. If so, please focus the main message of the manuscript on this fact, and highlight it in the Abstract section also. The authors mention four groups of animals, but only three are listed here.

3.     Abbreviations should be clarified at the first appearance (GC-MS, DPPH, ABTS, FRAP).

4.     In the Introduction section, the classification of diabetes requires amendment. According to the American Diabetes Association, diabetes is generally categorized into four groups, not types: Type 1, Type 2, gestational diabetes mellitus (GDM), and diabetes caused by other conditions.

5.     For this reviewer, it is evident from Table 1 and the provided information that there was no mortality during the study. However, please insert the number of animals excluded from the study in the 'Induction of Diabetes Mellitus' section. The fact that none of the animals died during the study raises this reviewer’s concern. Please explain the query that neither the streptozotocin treatment nor the extremely high blood glucose levels (600mg/dl) were lethal.

6.     In Table 2 the abbreviation IR is misspelled (RI), and in this reviewer’s version of the manuscript, there is no literary retention index listed.

7.     The abbreviation of the experimental groups should be consistent throughout the manuscript (e.g., Figure 3, 'control-system'). The grouping of the animals begs the question, why did the authors not create a healthy control group? According to this reviewer’s opinion, it is necessary to compare the effects of streptozotocin and the treatments with healthy animals, to be able to decide whether the model is satisfactory.

8.     Including a section to conclude the connection between reactive oxygen species (ROS) and metabolic changes in diabetes would enhance the quality of the manuscript, making the complexity of the disease more understandable for readers. This reviewer is curious about the author's opinion regarding whether the blood parameter changes are consequences of the antioxidant activity of lemongrass oil or if the antidiabetic effects caused a reduction in reactive species markers. Please, insert relevant citations from the literature to support this discussion.

Minor comments

1.     Please specify the number of the used laboratory animals in the “Animals” section first.

2.     In this reviewer’s version of the manuscript the letter size is different in the legend of Figure 2.

Comments on the Quality of English Language

Minor editing of English language needed. 

Author Response

- In the title of the article the “Rat Models” are in Plural Form, which indicates that there will be more than one model mentioned in the article. Since the authors describe only the streptozotocin-induced model of diabetes mellitus, please correct this grammatical mistake.

Thank you for your attention, the spelling was indeed inadequate, we have already changed the word to the singular.

- There are numerous mistakes in the Abstract section which indicates it to be rewritten. Based upon this reviewer’s understanding (according to the GC-MS examination’s results), the main active component of the EOCF is citral, which is responsible for the positive effects demonstrated in the Results part of the manuscript. If so, please focus the main message of the manuscript on this fact, and highlight it in the Abstract section also. The authors mention four groups of animals, but only three are listed here.

We apologize for the error previously written in this session. Thank you for your attention in the summary section, the previous writing did not justify that the control group received Tween 80 solution in its supplementation, nor did it specify that there were 2 groups that received OECF supplementation in two doses 32mg/kg and 64mg/kg. We made these adjustments. In relation to the questioning of the active component citral in EOCF, in the antioxidant activity tests OECF showed a higher antioxidant percentage than citral alone, as well as in the in vivo study where in some parameters EOCF stood out. The discussion and conclusion of the manuscript state that these results may be linked to the synergism between citral and the other components present in the EOCF.

- Abbreviations should be clarified at the first appearance (GC-MS, DPPH, ABTS, FRAP).

Thanks for the feedback, we've made the corrections and clarified the abbreviations in the first appearances.

GC-MS - line 27, DPPH - line 102, ABTS - line 108, FRAP - line 113.

- In the Introduction section, the classification of diabetes requires amendment. According to the American Diabetes Association, diabetes is generally categorized into four groups, not types: Type 1, Type 2, gestational diabetes mellitus (GDM), and diabetes caused by other conditions.

Thank you for your comment and we have changed the word to "categories" according to the American Diabetes Association. You can find the change on line 47.

- For this reviewer, it is evident from Table 1 and the provided information that there was no mortality during the study. However, please insert the number of animals excluded from the study in the 'Induction of Diabetes Mellitus' section. The fact that none of the animals died during the study raises this reviewer’s concern. Please explain the query that neither the streptozotocin treatment nor the extremely high blood glucose levels (600mg/dl) were lethal.

Thank you for your concern about the survival of the animals during the experiment. During the induction of DM, twenty-five animals were used, two of which did not resist STZ and three of which did not reach the blood glucose levels to be considered diabetic (150mg/dL) and were excluded from the study as described in line 135/136. As a result, only 20 animals were randomized to begin supplementation.  The text has therefore been rewritten to make it easier for the reader to understand on line 38.

- In Table 2 the abbreviation IR is misspelled (RI), and in this reviewer’s version of the manuscript, there is no literary retention index listed.

We apologize for the mistake, we have already rewritten the correct acronym in the text. Regarding the literary retention list, we apologize for the divergence between what was described in the subtitle and what was represented in the table. We have already made the change and it can be found updated in the manuscript. We would also like to inform you that the data has been inserted considering the literary indexes described according to the literature, to make it easier for the reader to understand in the methodology section on line 101. We would also like to inform you that the acronym traits has been removed from the table as no trait components were found in the analysis.

- The abbreviation of the experimental groups should be consistent throughout the manuscript (e.g., Figure 3, 'control-system'). The grouping of the animals begs the question, why did the authors not create a healthy control group? According to this reviewer’s opinion, it is necessary to compare the effects of streptozotocin and the treatments with healthy animals, to be able to decide whether the model is satisfactory.

I would like to clarify the abbreviations. In our manuscript, in vitro (antioxidant) tests were carried out with the control system as shown in figure 01. Figure 02 shows in vivo experimental analyses in which the control group received supplementation containing Tween 80 (TW 80).

With regard to the reviewer's concern about the satisfaction of the experimental model, we stress that the use of healthy animals in this experiment is not necessary, since this model is well established in the literature and is widely used by research groups at the university where the experiment took place.

da Silva, D. H. A.; Barbosa, H. M.; da Silva, J. F.; Moura, C. A.; Gomes, D. A.; Almeida, J. R. G. S.; Lira, E. C. Antidiabetic Properties of Oral Treatment of Hexane and Chloroform Fractions of Morus Nigra Leaves in Streptozotocin-Induced Rats. An. Acad. Bras. Ciênc. 2021, 93, e20210744. https://doi.org/10.1590/0001-3765202120210744.

Barman, S.; Srinivasan, K. Ameliorative Effect of Zinc Supplementation on Compromised Small Intestinal Health in Streptozotocin-Induced Diabetic Rats. Chem. Biol. Interact. 2019, 307, 37–50. https://doi.org/10.1016/j.cbi.2019.04.018.

Falode, J. A.; Olofinlade, T. B.; Fayeun, G. S.; Adeoye, A. O.; Bamisaye, F. A.; Ajuwon, O. R.; Obafemi, T. O. Free and Bound Phenols from Cymbopogon Citratus Mitigated Hepatocellular Injury in Streptozotocin-Induced Type 1 Diabetic Male Rats via Decrease in Oxidative Stress, Inflammation, and Other Risk Markers. Pharmacol. Res. - Mod. Chin. Med. 2023, 7, 100234. https://doi.org/10.1016/j.prmcm.2023.100234.

Mishra, C.; Khalid, M. A.; Nazmin, F.; Singh, B.; Tripathi, D.; Waseem, M.; Mahdi, A. A. Effects of Citral on Oxidative Stress and Hepatic Key Enzymes of Glucose Metabolism in Streptozotocin/High-Fat-Diet Induced Diabetic Dyslipidemic Rats. Iran. J. Basic Med. Sci. 2019, 22 (1), 49–57. https://doi.org/10.22038/ijbms.2018.26889.6574.

Akinlade, O. M.; Owoyele, B. V.; Soladoye, A. O. Streptozotocin-Induced Type 1 and 2 Diabetes in Rodents: A Model for Studying Diabetic Cardiac Autonomic Neuropathy. Afr. Health Sci. 2021, 21 (2), 719–727. https://doi.org/10.4314/ahs.v21i2.30.

We would also like to point out that the animals' blood glucose levels were assessed and only animals with blood glucose levels of 150mg/dL were included in the study.

-  Including a section to conclude the connection between reactive oxygen species (ROS) and metabolic changes in diabetes would enhance the quality of the manuscript, making the complexity of the disease more understandable for readers. This reviewer is curious about the author's opinion regarding whether the blood parameter changes are consequences of the antioxidant activity of lemongrass oil or if the antidiabetic effects caused a reduction in reactive species markers. Please, insert relevant citations from the literature to support this discussion.

Thank you for your analysis, it really is essential for the reader's better understanding. We have added another paragraph of discussion to the manuscript, which can be found on lines 350-356. We have also clarified the question "authors' opinion".

-  Please specify the number of the used laboratory animals in the “Animals” section first.

Thank you for identifying this difference between the font sizes, we apologize and have already made the change.

-  In this reviewer’s version of the manuscript the letter size is different in the legend of Figure 2.

Thank you very much, the number of animals in the experiment has been entered.

Reviewer 2 Report

Comments and Suggestions for Authors

Authors of the manuscript entitled “Influence of Lemongrass Essential Oil (Cymbopogon flexuosus) supplementation on Diabetes in Rat Models”investigate the antioxidant, hypoglycemic, anti-dyslipidemic and oxidative stress effects of EOCF in Wistar rats induced with streptozotocin (STZ) to develop DM1. but some points required more clarification by the authors

1.      The authors are advised to provide description about the positive control group (Trolox group) within the antioxidant assay, description should be added to section 2.6. Experimental Groups.

2.      Regarding the hyperlipidemic and hyperglycemia assays, the authors did not uses any positive control. Using positive reference group is highly advised to ensure the clinical significance of the biological results.

3.      Please provide rational for using double doses of EOCF instead of wider range of concentration.

4.      In table 2, there are missing column for literary retention index, tr: traits which mentioned in the table ligand.

5.      Please provide a column for references studies from which the literature data were compared.

6.      The authors are advised to represent the structures of the identified compounds.

7.      The authors should elaborate more on the safety of the EOCF since, the EOCF groups showed animal death as compared to citral. This is quite contradictory with the conclusion the authors raised within the conclusion section.

Author Response

-  The authors are advised to provide description about the positive control group (Trolox group) within the antioxidant assay, description should be added to section 2.6. Experimental Groups.

Thank you for your comment. However, the trolox group is used in in vitro antioxidant analysis (topic 2.3) and does not fit into topic 2.6. The group has been detailed in topic 2.3.

-  Regarding the hyperlipidemic and hyperglycemia assays, the authors did not uses any positive control. Using positive reference group is highly advised to ensure the clinical significance of the biological results.

We appreciate the suggestion and it will certainly be taken on board by our research group in future studies. For this research, we took into account Normative Resolution No. 12, September 20, 2013 of the National Council for the Control of Animal Experimentation (CONCEA) and the minimum number of animals necessary to achieve the scientific objectives. Thus, one of the requests of the Ethics and Experimental Research Committee (CEU) of the Federal University of Sergipe is respect for the Russell-Burch Principles (1959) of "reduction, substitution and refinement" in the use of animals, known as the 3R's Principle.

-  Please provide rational for using double doses of EOCF instead of wider range of concentration.

We thank you for your concern and inform you that in an in-depth and detailed analysis of the safe intake of other essential oils as well as the isolated compound citral orally, the literature recommends a dose of 32mg/kg, taken daily at the same time via the intragastric route.

Therefore, we used the dose suggested in the literature and double the dose to check whether increasing the dose would cause significant differences when compared to the recommended dose. As we think this information is relevant, we have a reference to the dose in the Supplementation section on line 164.

-  In table 2, there are missing column for literary retention index, tr: traits which mentioned in the table ligand.

Regarding the literary retention list, we apologize for the divergence between what was described in the subtitle and what was represented in the table, we have already made the change and it can be found updated in the manuscript. We would also like to inform you that the data has been entered taking into account the literary indexes described in the literature, to make it easier for the reader to understand in the methodology section on line 101. We would also like to inform you that the acronym traits has been removed from the table as no trait components were found in the analysis.

-  Please provide a column for references studies from which the literature data were compared.

Thank you, as mentioned in the previous suggestion, we have made the change and it can be found updated in the manuscript.

-  The authors are advised to represent the structures of the identified compounds.

-  The authors should elaborate more on the safety of the EOCF since, the EOCF groups showed animal death as compared to citral. This is quite contradictory with the conclusion the authors raised within the conclusion section.

We thank you for your concern regarding the survival of the animals during the experiment and apologize for the typing error regarding the animals. We would therefore like to clarify that during the induction of DM, twenty-five animals were used, two of which did not resist STZ and three of which did not reach the blood glucose levels to be considered diabetic (150mg/dL) and were excluded from the study as described in line 135/136. As a result, only 20 animals were randomly assigned to begin supplementation, with no deaths occurring during the 14 days of treatment. Therefore, the text was rewritten to make it easier for the reader to understand on line 38 and table 01 was updated.

Reviewer 3 Report

Comments and Suggestions for Authors

In this study the authors investigated the the impact of essential oil of Cymbopogon flexuosus, and its major component, citral, on glycemic, lipid, antioxidant parameters, and oxidative stress in a type 1 diabetes rat model. The results showed that Cymbopogon flexuosus demonstrated efficacy in ameliorating chronic metabolic alterations associated with type 1 diabetes.

The level of originality of the paper is high. This study can be very useful for the prospect of oral lemongrass essential oil supplementation in DM1 treatment. Current literature is properly discussed and compared to the previous studies. The introduction needs to be improved.

Specific comments:

Line 70: „However, studies using the essential oil of Cymbopogon flexuosus (EOCF) in this area are scarce in the literature“. Please include references for this sentence. Maybe add what has already been researched and what is new in your study.

Line 100: Determination of antioxidant activity: The method that you used for antioxidant activity are not properly described and they can hardly be repeated for readers. Why did you used Trolox insted DPPH in DPPH assay?

Figure 1: What is the meaning of lower case on the figure? I don't understand how did you gave the lowercase letters, should lowercase letter b be written instead of c, and c should be written instead of b?

Line 224: „The ABTS radical scavenging method is similar to DPPH.“ on what basis do you claim that the two methods are similar? The obtained results are not the same. Did you perform a statistical analysis between the methods?

Figure 2, 3 and 4: There is no lowercase letter on the figures obtained by Tukey post-test. Please add lowercase letters to the figures.

Author Response

-  Line 70: „However, studies using the essential oil of Cymbopogon flexuosus (EOCF) in this area are scarce in the literature“. Please include references for this sentence. Maybe add what has already been researched and what is new in your study.

Thank you for your contribution, it makes perfect sense given that the studies that use the types of Cymbopogon do not show flexuosus for the treatment of DM or insulin resistance. The literature mainly uses the citratus species and others such as proximus, jwarancusa and martinii. We have added to the end of the statement in line 73 studies using the species mentioned.

-  Line 100: Determination of antioxidant activity: The method that you used for antioxidant activity are not properly described and they can hardly be repeated for readers. Why did you used Trolox insted DPPH in DPPH assay?

Thank you for your contribution. Improvements have been made to the description of the methodology to make it easier to replicate. Regarding trolox in the DPPH reduction method, it was used as an antioxidant standard.

-  Figure 1: What is the meaning of lower case on the figure? I don't understand how did you gave the lowercase letters, should lowercase letter b be written instead of c, and c should be written instead of b?

Thank you for your question. Detailed information on the meaning of the letters and asterisks has been added. In summary, the letters indicate a difference only between the samples tested (EOCF, Citral and Trolox). The asterisks indicate a difference between the samples and the system (no antioxidant).

-  Line 224: „The ABTS radical scavenging method is similar to DPPH.“ on what basis do you claim that the two methods are similar? The obtained results are not the same. Did you perform a statistical analysis between the methods?

Thank you for your question. In fact, the methods are similar only in that they use synthetic radicals. There are important differences: the DPPH reduction method is based only on hydrogen donation, and only polar substances are easier. The ABTS uptake method evaluates hydrogen and electron donation, and both polar and apolar substances can interact. No statistical analysis was carried out between the methods. The text indicating similarity between the methods was adapted to avoid misinterpretation.

-  Figure 2, 3 and 4: There is no lowercase letter on the figures obtained by Tukey post-test. Please add lowercase letters to the figures.

Thank you for your suggestion, however, I would like to explain that the asterisk represented by its symbol (*) already shows the significant difference between the group in question and the control group, while the antiphenon (#) expresses the significant difference between the control group and the group represented by the *.

Round 2

Reviewer 1 Report

Comments and Suggestions for Authors

In this article Ailton S. S. Júnior, and colleagues describe the clinical significance of the essential oil content of lemongrass in relation to type 1 diabetes mellitus and its potential effects on lipid homeostasis. The overall topic is intriguing, and this manuscript aims to expand opportunities in the treatment of diabetes mellitus, a global health challenge.

The authors corrected all of the mistakes this reviewer highlighted, and enlightened the misunderstandings, by the complementation of the manuscript it reached a high quality, and provides useful information for patients and clinicians as well, which makes it acceptable for publication.

Comments on the Quality of English Language

fine

Reviewer 2 Report

Comments and Suggestions for Authors

The authors of the manuscript responded to almost all comments, but some points required more clarification by the authors

·         The authors are advised to provide description about the positive control group (Trolox group) within the antioxidant assay, description should be added to section 2.6. Experimental Groups.

The text added by the authors in section 3.2 is not relevant to posted comment. A positive control group is a group of mice that would administered a reference market or literature reported antidiabetic drugs. Comparing the authors result with those of positive control group will highlight the clinical significance of the extract and identified compounds in term of potency.

The authors mentioned inadequacy in referral to ethics committee, this should be added as limitations of this study.

·         Please provide rational for using double doses of EOCF instead of wider range of concentration.

The authors response and rational should be added to the manuscript.

·         In table 2, there are missing column for literary retention index, tr: traits which mentioned in the table ligand.

According to the response there are several columns added to the table but there are missing in the attached revised version. Authors should revise the table

·         The authors are advised to represent the structures of the identified compounds.

The authors did not reply to this comment.

Author Response

We sincerely thank the editor and the reviewers for their attention and dedication to our manuscript. Their constructive comments and suggestions have certainly made an important contribution to improving our manuscript. Please find below a point-by-point response to all the reviewers' comments and recommendations. The language of the article has been carefully revised to make it clearer and more precise. Additionally, we have added highlights to the text indicating the areas where changes were made as a result of the last revision.
